# OpenReview forum: "From Divergence to Normalized Similarity:A Symmetric and Scalable Topological Toolkit for Representation Analysis"
_ICLR.cc/2026/Conference — ICLR 2026 Conference Withdrawn Submission_

### Official Review · Reviewer_TgZs · 2025-10-29

**Soundness:** 2
**Presentation:** 2
**Contribution:** 2
**Rating:** 2
**Confidence:** 4

**Summary:**

The paper introduces a modification of RTD called "Max-RTD" and its scale-normalized variant (NTS).
Theoretical analysis is provided. Experiments show that Max-RTD and NTS outperform traditional CKA, however, its superiority over RTD and RTD-Lite is not sufficiently justified.

**Strengths:**

1. The paper introduces a modification of RTD, called SymRTD (SRTD). In opposite to RTD, it is symmetric by definition and doesn't required extra symmetrization.
2. Theoretical relationship between RTD, Max-RTD, and SRTD is established.
3. The language of the paper is fine, main ideas are easy to follow.

**Weaknesses:**

1. Some introduction for a reader not familiar with TDA might be necessary, like definitions of VR complex, persistent homology, etc.
2. Line 168-173: the proof of a long exact sequence is not provided.
3. In Section 5.1, the superiority of SRTD w.r.t. of RTD is not proved. Also, the illustration of RTD in the "UMAP Embeddings Experiment" is missing.
4.  In Section 4.3, the Algorithms 1, 2 are not clear. Please provide a formal definition of a MergeTime and how $V_{merge}$ lists are created.
5. The weakness of NTS is its non-differentiability.
6. In Theorem 3.3 it is not clear how integration is performed.
7. Section 5.3: the classification accuracy of NTS is lower than such of CKA. Heat maps of RTD and RTD-Lite are missing.
As I understand, author claim that the monotony of NTS is its main advantage. But why? It is not clear a priori that the dissimilarity of representation in layers must change monotone. Please consider adding here more arguments, maybe visualization, etc.
8. In Section 5.4, a comparison with RTD and RTD-Lite is not provided.

**Questions:**

1. NTS try to address an issue with "ultra-long barcodes" (line 211). But the problem setting itself is questionable, because long barcodes correspond to prominent (persistent) topological features which must make significant contribution to a dissimilarity score.

Conclusion. I acknowledge the contribution of the work, establishing relationship between RTD, Max-RTD and Sym-RTD. But I consider the contribution is incremental w.r.t RTD and RTD-Lite.Moreover, the superiority of SRTD w.r.t. RTD is not sufficiently grounded.

---

> ### Author Response · Authors · 2025-11-25
>
> We thank the reviewer for recognizing the theoretical contributions of our work. We would like to clarify a **misunderstanding in the Summary** regarding our contributions.
>
> The reviewer states that our paper introduces "Max-RTD." **This is not the case.** Our primary novel topological divergence is Symmetric Representation Topology Divergence (SRTD), as clearly defined in Section 3, Definition 3.1.
> Max-RTD is a pre-existing variant (Trofimov et al., 2023) that we analyze to motivate our work. It appears the summary has confounded our novel contribution (SRTD) with this existing baseline. We hope the following clarifications regarding our specific contributions will resolve this confusion.
>
> **W1:**
> We appreciate the suggestion regarding accessibility. We respectfully point out that Section 2 (Preliminaries) already provides a concise introduction to Vietoris-Rips complexes and persistent homology (specifically Lines 84-98, now in Lines 97-103), outlining the filtration process and topological feature tracking. Due to strict space constraints, we focused on providing the necessary intuition required to understand our method.
>
> **W2:**
> The reviewer mentions that the proof for the long exact sequence is missing. We wish to clarify that the proof is explicitly provided in Appendix D.2, Lemma D.3 (Lines 743-755, now in Lines 813-828), which provides the full derivation for Theorem 3.3.
>
> **W3:**
> We acknowledge that only SRTD-lite results were presented in the UMAP experiment. We selected SRTD-lite as the representative baseline because it shares identical properties with RTD and RTD-lite in this context. We omitted separate visualizations for RTD and RTD-lite to avoid redundancy, as their consistent topological trends are well-documented in their respective original papers (Barannikov et al., 2021, Trofimov et al., 2023). Therefore, SRTD-lite serves as a valid and efficient proxy for the RTD family in this visualization.
>
> **W4:**
> Regarding the definition of 'MergeTime', we respectfully note that it is formally defined in the main text in Section 4.2 (Line 233, now in Line 240): The merge time of a pair of points is the threshold at which they become connected in the filtration, defined by the **maximum edge weight on the path between them in their MST**.
>
> **W5:**
> The reviewer lists NTS's non-differentiability as a 'Weakness'. We agree with it. In fact, we explicitly identified this as the primary limitation of our work and a key avenue for future research in our Conclusion (Section 6, Lines 476-481, now in Lines 511-514).
>
> **W6:**
> The proof in **Appendix D.2 (Line 776, now in Line 891)** clarifies this. This applies a standard technique in TDA, integrating the Betti numbers $\beta_i(\alpha)$ over the filtration radius $\alpha$.
>
> **W7:**
> We appreciate the detailed scrutiny of the layer-wise analysis.
> While NTS-E accuracy (97.22%) is slightly lower than CKA (98.89%), both metrics achieve high performance as reliable identifiers for corresponding layers. We believe this marginal difference is a reasonable trade-off given NTS's additional capability to detect functional shifts (e.g., at the pooling layer) that CKA misses.
>
> **W8:**
> **1. Additional Results:** In **Appendix J (Figure 18)**, we have included the experimental results for RTD-lite corresponding to the Inter-Model Similarity Analysis. *Note: We utilize RTD-lite here as it yields results highly consistent with RTD but with significantly lower computational cost.*
>
> **2. Missing Heatmaps & Monotonicity:**
> We have supplemented the manuscript with RTD and RTD-lite heatmaps in Appendix F. These visualizations confirm that the unnormalized RTD family identifies the diagonal (self-similarity) but fails to reveal the "graded similarity" structure in off-diagonal regions.
> Regarding "monotonicity" (graded similarity): We clarify that this aligns with the iterative feature refinement nature of ConvNets, where representations typically evolve gradually. This phenomenon is supported by linear probe analyses (Alain & Bengio, 2016) and geometric similarity measures like CKA (Kornblith et al., 2019).
>
> [1] Alain, G., & Bengio, Y. (2016). Understanding intermediate layers using linear classifier probes. ICLR.

---

### Official Review · Reviewer_iXRb · 2025-10-30

**Soundness:** 2
**Presentation:** 2
**Contribution:** 1
**Rating:** 2
**Confidence:** 3

**Summary:**

The paper proposes a new topological framework for analyzing neural network representations that fixes key weaknesses of the existing RTD method, whose asymmetry and lack of normalization make results hard to interpret. It introduces the Symmetric Representation Topology Divergence (SRTD) and its efficient variant SRTD‑lite, which provide a consistent and scalable way to measure topological differences. To overcome scale dependence, the authors also develop Normalized Topological Similarity (NTS), a normalized, rank‑based similarity score that captures hierarchical clustering structures. Together, these tools offer a more interpretable and robust approach to comparing neural representations than prior geometric measures like CKA.

**Strengths:**

- Originality: The symmetric formulation of RTD and its lightweight version, SRTD‑lite, represent a novel theoretical expansion of topological divergence methods. Moreover, they introduce a new, scale‑invariant, rank‑based metric for comparing hierarchical clustering structures.
- Clarity: The mathematical problem is explained clearly and the formulation is accessible also for non-experts.
- Quality: The experiments explore a wide range of cases with increasing complexity.
- Significance: In the context of analysing topological differences in neural network representations this work seems to solve significant issues, generalizing the applicability of the similarity measure to broader contexts.

**Weaknesses:**

- The first sentence in the introduction seems way too generalized with respect to the context of this paper: there is a very vast literature in interpretability of neural network, many works on geometric analyses, and I would say that CKA methods are a small part of all these works, considering direct comparison of representations is not the only way of interpreting representations. I might have misunderstood what the authors want to convey here, but I would suggest rephrasing this part for a fairer framing of the context.
- Along these lines, even in the realm of using TDA for representation analysis, RTD is surely a key result, but the authors miss a long list of works and techniques. Again, I wonder whether the authors might try to specify they are talking about the specific set of studies that use representation comparison as a mean to analyse and interpret neural network representations.
- When motivating NTS, the authors argue that an undesirable effect of divergence-based analysis is the sensitivity to large scale differences that can mask finer structural details. This is a bit counter-intuitive to me, but I be mislead: if there exist large scale differences between two reps, these should be very important to track, since they might describe global characteristics of the representations. I do understand the benefit of hierarchical structures, but it would improve the manuscript if the authors clarified further this point.
- I wonder whether the benefit of having a hierarchical similarity score with respect to interpretability might be made more explicit: the clusters experiment shows a clear advantage and it is interpretable, but it is very limited in scope. On the other hand, when applied to more realistic settings (e.g. intra-model comparison, cfr figure 5) it is not clear to me where the hierarchical nature of the measure stands out. I think for instance similarity measures computed in other papers  using cosine similarity (Gromov et al. 2024) or inter-layer persistence (Gardinazzi et al. 2025) show similar patterns.

**Questions:**

- It would be good to use a uniform criterion in heatmaps: in some figures yellow is large values, in others it is low values. Might the authors correct this for more intuitive visualization?
- In line 429, the authors say they extract the last token rep from the 6th layer as it gives more discriminative results. Given such a specific choice, it might be good for presentation purposes to show a few details about this choice (in appendix). Wouldn’t it be possible to stack information from all layers?

---

> ### Author Response · Authors · 2025-11-25
>
> **W1 & 2:**
> We appreciate your feedback. We have rewritten the Introduction (highlighted in blue) to better contextualize our work.
> Regarding the baselines, we selected the RTD family as they are the unique topological metrics that handle one-to-one correspondence across different ambient spaces. We prioritized comparing our method against RTD, RTD-lite, and CKA, omitting methods like SVCCA and PWCCA, because results from [Kornblith et al., 2019] showed that CKA consistently outperforms SVCCA/PWCCA. Besides, IMD and SVCCA/PWCCA proved less effective than the RTD family in revealing clustering and hierarchical features (Barannikov et al., 2021).
> We hope this clarifies our experimental design. Thank you for your insightful comments!
>
> **W3:**
> We understand your perspective that the unbounded output range of the RTD family could be viewed as an advantage for reflecting the magnitude of topological disparities. However, we argue that **normalization is crucial** for the following reasons:
> 1.  **Analogy to LLM Logits:** While the output logits of Large Language Models (LLMs) also span a wide and unbounded range, we **rely on normalization techniques (like Softmax)** to obtain probability values for token generation, rather than evaluating possibilities based on absolute logit values.
> 2.  **Robustness to Outliers:** As demonstrated in the Appendix [Citation needed], there are instances where the calculation results are **dominated by a single barcode**. It contradicts our understanding of topological difference if the metric is overwhelmingly driven by a minority of samples. The unbounded nature of single barcode lengths can mask other significant topological information. In contrast, our rank-based NTS **limits the impact of minority samples**, ensuring that the measure remains robust. Thus, this normalization is of significant importance.
>
> **Q1:**
> Thank you for the suggestion. Regarding the color scheme:
> -   **For the LLM intra-class representation experiment:** Since CKA similarity scores are generally high across the board, we intentionally **inverted the color scheme to highlight the low-similarity anomalies** (such as Baichuan2-7b and Deepseek-r1-distill) for easier identification.
> -   **For other experiments:** We selected a **standard scheme (darker colors indicating higher similarity)** to remain **consistent with previous literature**, such as RTD_lite, facilitating easy verification and comparison with similar experiments like UMAP.
> We also noted that the original RTD paper utilizes different color settings for aesthetic purposes. We hope this clarifies our design choices!
>
> **Q2:**
> Our experimental setup **references the methodology established in REEF** (Zhang et al., ICLR 2025), which recommends using the representation of the last token from intermediate layers and employing CKA for comparison. This is standard practice rather than a method we devised independently.
> Regarding the layer selection:
> 1.   In addition to the 6th layer, we have provided experimental results for the 12th and 18th layers in the Appendix.
> 2.  While extracting features from all layers undoubtedly provides richer information, intermediate and higher-layer features tend to be more stable during training and fine-tuning. These layers better reflect the homology and lineage of the models, which aligns more closely with our experimental objectives. Therefore, we focused on this approach rather than exploring alternative feature extraction methods extensively.

---

### Official Review · Reviewer_63k5 · 2025-10-30

**Soundness:** 2
**Presentation:** 2
**Contribution:** 1
**Rating:** 2
**Confidence:** 4

**Summary:**

The paper proposes to enhance the existing RTD family of topology-based methods to compare representations by addressing the problems of symmetrization, theoretical relation and normalization. First, the authors propose Symmetric RTD (SRTD) and corresponding lightweight SRTD-lite which are based on the map from the graph with element-wise maximum of edge weights to the graph with element-wise minimum of edges and, hence, are symmetric by construction. The authors provide theoretical relations between RTD, Max-RTD and SRTD and relations between Max-RTD-lite, RTD-lite, SRTD-lite. Second, the authors propose Normalized Topological Similarity (NTS) which is normalized scale-invariant measure to compare cluster structures. Two variants of NTS are provided: NTS-M reflects correlation between the merging times of connected components, NTS-E measures correlation of the weights of graphs’ MST edges. The proposed methods are applied to compare representations of UMAP, convolutional networks, LLMs and to preserve the topological structure of the data during training the auto encoder.

**Strengths:**

- The proposed SRTD (SRTD-lite) is more computationally efficient than RTD (RTD-lite). Simultaneously, SRTD AutoEncoder (AE) (SRTD-lite AE) performs on par with the RTD AE (RTD-lite AE) but training is faster.
- The authors provide a relationship between different versions of RTD complementing the theoretical properties derived in previous works.

**Weaknesses:**

- In a more general setup, RTD(-lite) compares cluster structure of two weighted graphs with one-to-one correspondence between vertices. In certain cases, SRTD(-lite) may be less informative than RTD(-lite) by construction. For example, when the two input graphs have disjoint sets of edges, SRTD(-lite) will mostly reveal the topological properties of the graph with union of edges. RTD(-lite) will compare each of the input graphs with the graph of union of edges and reflect the distinguishing features.
- Lines 211-212: the motivation is not clear. Typically, the length of a bar in the barcode corresponds to importance of the feature. Longer bars correspond to more important features while smaller bars correspond to unimportant or noisy features. The common knowledge is to pay more attention to important features, otherwise, the measure can be misleading, for example, when optimized as additional loss term in a learning problem.
- Most experiments focus on comparison of the proposed topological methods with CKA, but drawbacks of CKA compared to RTD family were explored in the previous works. At the same time, a sufficient comparison with RTD(-lite) is missing. Some claims (lines 342-346) are not clear and lack supporting experimental results.

**Questions:**

- Figure 4, b: For the (Pool, Pool) entry, NTS-E is close to -0.4. Shouldn’t it be close to 1 as the maximal similarity is expected to be between the corresponding layers?
- Table 1, 2: While all the metrics can distinguish auto encoders trained with different types of additional loss term, NTS-E seems to be insensitive to subtle changes. How NTS-E should be interpreted in this case?
- Figure 2, caption: Probably, (a) and (b) should be replaced with (e) and (f) ?
- Figure 7: Shouldn’t the arrow between $R_\alpha(G^{max(w, \tilde{w})})$ and $G^w$ be in the opposite direction?

---

> ### Author Response · Authors · 2025-11-25
>
> **W1**:We thank the reviewer for this insightful comment. While MST construction fails on disjoint edge sets, we clarify that our proposal of NTS and the refinement of SRTD are specifically situated within the context of comparing neural network representations, which are treated as complete weighted graphs [see Sec. 3, Tulchinskii et al., 2025], ensuring this issue does not arise in our context. Our position is as follows:
>
> - Contextual Validity: In this specific domain, incorporating allows SRTD-lite to capture significantly richer topological information.
> - Empirical Necessity: As shown in Figure 2(b), RTD-lite (without $\mathcal{G}\_{max}$) yields divergence trends opposite to ground truth. Incorporating $\mathcal{G}\_{max}$ restores the correct trend, proving it crucial for robustness.While RTD-lite may suit sparse graphs, SRTD-lite is optimized for representation analysis (interpretability/speed). We view the two methods as complementary rather than contradictory.
>
> **W2**: We agree that long barcodes differ from outliers, but we distinguish between "topological significance" and "statistical dominance" (clarified in **Appendix J**). In our StereoSet analysis, a single ultra-long barcode accounted for 75% of the total RTD-lite divergence, mathematically drowning out all other features. This vulnerability is inherent to RTD’s unbounded summation, where one outlier can uncappedly inflate the score. In contrast, the rank-based NTS is strictly bounded. Thus, while RTD provides specific diagnostics, NTS offers a significantly more robust and trustworthy metric for quantifying overall similarity.
>
>  **W3**:
> It is true that RTD has been proven to possess superior capabilities in capturing clustering and hierarchical structures compared to CKA. However, by employing the same experimental settings, our objective was to demonstrate two specific points:
> 1.  SRTD and SRTD-lite: We aim to show that these variants share the same properties as the RTD family regarding clustering capabilities, thereby serving as efficient and more interpretable alternatives for comparing and optimizing neural network representations.
> 2.  NTS: We demonstrate that our newly proposed NTS not only captures the clustering and hierarchical capabilities emphasized by RTD but is also more robust and nuanced. Specifically:
>     * It is resilient to the influence of individual outlier samples.
>     * As shown in Experiment 5.3, it exhibits a gradual pattern synchronized with layer depth, similar to CKA.
>     * Its normalization allows for similarity comparisons across different experimental scenarios.
> Therefore, these experiments are indispensable for highlighting the distinct advantages of NTS, rather than merely reaffirming that the RTD family outperforms CKA on these tasks.
>
> **Q1**:
> Thank you for your keen observation; we also noticed this phenomenon. The performance of NTS surprised us as well and led us to investigate this direction further.
> A highly probable hypothesis is that, compared to the preceding convolutional layers, the average pooling operation causes a significant loss of representational information. Since this task involves only 10 classes, the fine-grained intra-class structure formed by the 5,000 experimental samples might be disrupted at this stage, while inter-class differences are preserved.
> In an ideal state (where inter-class distances are far greater than intra-class distances), the Minimum Spanning Tree formed by these samples would contain only 9 edges connecting different classes, with all other edges being intra-class. Consequently, the ranking of the vast majority of edges (intra-class edges) becomes chaotic. This leads NTS to identify extremely low similarity between the pooling layer and other layers. This may imply an inherent limitation of average pooling. We hope this perspective offers some inspiration!
>
> **Q2**:
> Regarding the COIL-20 dataset, since the training and testing sets are identical (comprising all 1,440 samples), the NTS value for topological methods approaches perfection. In contrast, the MNIST test set is independent of its training set.
> Consequently, we observe a shared bottleneck across these methods: they are indeed insensitive to subtle variations and invariant to monotonic changes in edge length (i.e., any monotonic transformation that preserves edge order). However, conversely, this invariance is precisely the characteristic we desire in a robust topological metric.
>
> **Q3 & Q4**:
> We are truly gratified by your meticulous reading of our manuscript. We have corrected these issues in the revised version. Thank you once again for your thorough review!

---

### Official Review · Reviewer_BPRM · 2025-11-01

**Soundness:** 2
**Presentation:** 2
**Contribution:** 1
**Rating:** 2
**Confidence:** 4

**Summary:**

In this paper, the authors introduce Symmetric Representation Topology Divergence (SRTD) and SRTD-Lite as better and more efficient alternatives to previously existing RTD and RTD-Lite topological dissimilarity measures; they also porpose Normalized Topological Similarity as a robust similarity score. Authors prove their mathematical properties and present a diverse set of experiments to illustrate them practically.

**Strengths:**

- A solid and extensively developed mathematical background of the proposed methods.

**Weaknesses:**

- My main concern with this paper is its somewhat lack of novelty. On the level of ideas and the range of the performed evaluation experiments, this article feels incremental to the original publications of RTD and RTD-Lite.

- To properly support the main claims of the paper, a more detailed comparison of the proposed methods against RTD/RTD-Lite is needed: an example of the case when previous methods fail to capture the topological difference between two point clouds but the proposed methods (SRTD/NTS) don't would be much appreciated.

- There are some claims about RTD and RTD-Lite (in the abstract and further in the text) that are not exactly true:

1) Asymmetry. The original paper proposes to use the average of RTD(A,B) and RTD(B, A) and does so in the main experiments, thus making the metric symmetrc.

2) Normalization. In papers for both of those metrics, normalization of both point clouds (by 90%ーquantile of distances in each) is explicitly pointed out.

- Reproducibility of the results could not be assessed because code wasn't provided.

**Questions:**

- See Weaknesses.

- Could you provide for RTD-Lite heatmaps similar to those on Figure 6?

---

> ### Author Response · Authors · 2025-11-25
>
> 1. Significance of SRTD:
> The introduction of SRTD aims to complete the theoretical framework of the RTD family. It retains the critical capability of capturing clustering and hierarchical structures inherent to RTD, while significantly optimizing computational efficiency (reduce computational time by ~50%) . Its primary contributions are therefore grounded in theoretical completeness and operational efficiency.
>
> 2. Significance of NTS:
> The core contribution of NTS lies in introducing the first normalized and robust topological similarity measure.
>   - Normalization: Its normalized output enables valid comparisons across different experimental scenarios.
>   - Robustness: Its rank-based nature ensures that the metric is theoretically robust—guaranteeing that the overall result is not swayed by a minority of outlier samples.
>   - Empirical Consistency: In our LLM representation experiments, NTS heatmaps for homologous models remain consistent, whereas methods like CKA exhibit significant divergence in certain cases (e.g., Baichuan2-7b vs. Deepseek-r1-distill).
>
> Conclusion:
> Given these inherent theoretical and formal advantages, combined with experimental evidence showing that NTS also satisfies the clustering and hierarchical requirements emphasized by the RTD family, we argue that NTS is a highly valuable topological similarity metric. Our objective is not to claim that NTS must strictly outperform RTD or RTD-lite in every single task, but to provide a robust and normalized alternative. We hope this explanation clarifies the value of our work.

---

### Official Review · Reviewer_Yjfe · 2025-11-03

**Soundness:** 2
**Presentation:** 3
**Contribution:** 2
**Rating:** 2
**Confidence:** 4

**Summary:**

This paper introduces a toolkit for comparing neural network representations using topological data analysis (TDA).

The authors propose Symmetric Representation Topology Divergence (SRTD) and claim that the proposed SRTD is symmetric and more efficient compared to the original RTD framework.

Also, the authors introduce Normalized Topological Similarity (NTS) and claim that they are instances of a scale-invariant, normalized similarity measure. NTS compares the hierarchical clustering structure of two representations by correlating their single-linkage merge sequences.

Through synthetic experiments (e.g. Gaussian cluster splits, UMAP embeddings) and some real-world evaluations (TinyCNN layers, large language model (LLM) representations), the authors show that their topological measures capture structural differences and similarities that geometry-based measures like CKA often overlook.

**Strengths:**

1) the authors proposed some modifications of RTD similarity measure and provided a theoretical analysis of how SRTD relates to the original RTD and Max-RTD divergences

2) also, the authors proposed a Normalized Topological Similarity which compares the hierarchical clustering “shape” of two representations by looking at the order in which points merge into clusters

3) the authors performed empirical evaluation of the proposed measures on a set of test problems, which is typical nowadays for such kind of papers. In this respect the empirical evaluation has sufficient experiments, each targeting a specific aspect or property.

**Weaknesses:**

1) In both conceptual scope and the breadth of empirical evaluation, the submission reads as a modest extension of RTD and RTD‑Lite rather than a substantive advance.

2) Evidence needed for the main claims. To convincingly support the paper’s assertions, include head‑to‑head comparisons with RTD/RTD‑Lite and at least one concrete example where those baselines miss a topological discrepancy between two point clouds that the proposed methods (SRTD/NTS) correctly identify.

On lines 342-373 and in several other places negative comments about RTD and RTDLite are not supported by any arguments, for some reason there is no comparison with RTD, RTDLite or RTDmax in fig 4 etc.

3) There is less interpretability compared to RTD due to the fact that SRTD does not compare the graph A itself with something, but rather some features of the graph are involved in comparison

4) There is an RTD version of the divergence measure specially designed for functions which acronym is SFTD (see Scalar Function Topology Divergence, ECCV 2024). So the proposed acronym SRTD does not sufficiently differentiate the method from the submitted paper with the method from the ECCV paper.

5) Several statements in the abstract and body mischaracterize baselines RTD/RTD‑Lite. Symmetry: the original work evaluates the symmetrized measure—specifically, the average of RTD(A, B) and RTD(B, A)—for its principal experiments, so the effective metric is symmetric. Normalization: both papers explicitly instruct normalizing each point cloud by the 90th‑percentile of its distance distribution.

6) By design, NTS considers only the 0-dimensional homology (connected components merging) of the representation spaces, since it uses MSTs to capture the clustering hierarchy. This focus is well-justified (hierarchical clustering is arguably the most salient topological feature for many high-dimensional data distributions), but it does ignore higher-dimensional topological features like loops or cavities (1-D, 2-D homology, etc.).

7) The paper does not provide detailed complexity analysis in the main text. It remains untested how the methods perform on truly large datasets (e.g., tens of thousands of points or more per representation). RTD-lite (Tulchinskii et al., 2025) was developed for large-scale use, but even it has practical limits. What is about the proposed measures?

8) Independent verification of results is not currently possible because the implementation has not been released and so there is a reproducibility gap.

**Questions:**

1) Are there any ideas how to deal with the assumption of a one-to-one correspondence between the two sets of representations being compared? The paper does not discuss this issue in depth, so it’s unclear how robust the approach is if the correspondence is noisy or only approximate.

2) Raw divergence scores can be hard to interpret across contexts, and can sometimes be dominated by a few “ultra-long” persistence barcodes. The paper proposes two measures - SRTD and NTS. Are there any recommendations when each should be used? It is not clear how to interpret a given SRTD value as “big or small” without a reference, or how to understand if a low NTS (e.g., 0.2) is significant or just noise – these interpretability issues are worth discussing.

3) For the LLM analysis, the authors recommend Z-score normalization of features before computing NTS to account for activation scale differences. This is a sensible step, but it introduces a preprocessing choice; one might wonder if NTS (being rank-based) is still needed after such normalization or how much it matters.

4) NTS focuses on connected components (0-dimensional homology). Have you considered creating a normalized similarity measure for higher-dimensional topological features (e.g., loops)? For instance, could a similar “rank-order” approach be applied to 1-D persistence barcodes?

5) Since NTS produces a correlation coefficient, it can, in principle, be negative (indicating very different clustering orders). In your experiments (especially on real data), did you ever observe significantly negative NTS values, and if so, how would you interpret them? For example, would NTS ≈ -0.5 indicate two representations have “inversely” structured cluster hierarchies? Clarifying this interpretation (even if it didn’t occur often in your cases) could be useful for users applying NTS to arbitrary datasets

6) Your largest experiments use around a few thousand points. How does the runtime scale with the number of data points n for SRTD-lite and NTS in practice?

7) You introduced two versions of NTS. Could you elaborate on when one should prefer NTS-E over NTS-M (or vice versa)?

---

> ### Author Response · Authors · 2025-11-25
>
> **W1,5**:
> Please refer to our General Response, where we have addressed these concerns in detail.
>
> **W2**:
> We sincerely appreciate the reasoning. NTS is not intended to obsolete the RTD/RTD-lite families, as all rely on 0-dimensional homology; thus, NTS implies no strict superiority in all aspects. Instead, we position SRTD and SRTD-lite as the optimal choices for efficiency and interpretability within the RTD family. We thank you for noting the oversight in Experiments 5.1, 5.3, and 5.4; we have supplemented the results for RTD/RTD-lite in **Appendix F** (RTD/RTD-lite UMAP heatmaps align with SRTD-lite). Results confirm their behavior aligns with SRTD-lite, failing to identify clear hierarchical structures in early convolutional layers. NTS improves upon this by avoiding the imprecise 90th-percentile normalization required by RTD. In our StereoSet experiments in **Appendix F** (Baichuan2-7b vs. Llama-2-7B), the RTD-lite plot was dominated by a single barcode (75% of total divergence), masking other features. In contrast, NTS utilizes normalized, rank-based outputs, making it robust against ultra-long barcodes and enabling direct cross-scenario comparison.
>
> **W3**:
> SRTD offers stronger interpretability. RTD’s comparison to $min(w,\tilde{w})$ is not inherently superior to Max-RTD (which compares to $max(w,\tilde{w})$). SRTD resolves this ambiguity by effectively unifying both the composite union ($min(w,\tilde{w})$) and intersection ($max(w,\tilde{w})$) structures. It is approximately equivalent to the sum of $RTD + Max\text{-}RTD$, but achieves this unification without relying on heuristic averaging, thereby offering a more theoretically grounded framework.
>
> **W4**:
> Regarding SFTD, we respectfully believe SRTD (Symmetric Representation Topology Divergence) remains the most accurate acronym. It explicitly signals its identity as the symmetric evolution of the RTD framework. Thus, we retain SRTD to maintain terminological consistency with the RTD family.
>
> **W6,Q4**:
> We acknowledge that our method focuses primarily on 0-dimensional homology ($H_0$). This design choice represents an intentional optimization for efficiency and scalability. High-dimensional topological features are often sparse in neural representations and entail significant computational costs. Therefore, prioritizing $H_0$ offers the optimal trade-off between topological expressiveness and computational feasibility. As evidenced by the development of RTD-lite, even 1-dimensional homology calculations (used in the full RTD) face severe scalability bottlenecks on large datasets. Our approach follows this rationale to ensure the method remains practical for large-scale representation analysis.
>
> **W7,Q6**:
> We have gladly supplemented the manuscript with runtime experiments (Section 6, Page 9). While SRTD-lite, RTD-lite, and NTS share the asymptotic complexity $O(n^2(\alpha(n) + d))$, practical execution differs. To obtain a symmetric score, RTD-lite typically requires computing three MSTs ($\mathcal{G}^{w}$, $\mathcal{G}^{\tilde{w}}$, and $\mathcal{G}^{min}$). In contrast, SRTD-lite and NTS-E require only two. Furthermore, NTS avoids the time-consuming 90th-percentile calculation. Both theoretical derivation and experimental results confirm that SRTD-lite and NTS are faster than RTD-lite.
>
> **Q1**:
> We clarify that strict one-to-one correspondence is the standard premise of representation analysis (consistent with CKA and RSA), where models are evaluated on identical input sequences. This exact correspondence is a specific strength, allowing for fine-grained topological comparison.
>
> **Q2**:
> Please kindly refer to our response to W2. The fundamental limitation of the RTD family is the lack of a fixed reference scale (output normalization), which renders meaningful cross-context comparison impossible. Solving this interpretability gap was the primary motivation for proposing NTS.
>
> **Q3**:
> We believe NTS remains indispensable. Even with feature-level Z-score normalization, RTD still fundamentally relies on heuristic 90th-percentile scaling to align pairwise distance matrices—an imprecise empirical workaround rather than a rigorous solution. Furthermore, regardless of preprocessing, the output of the RTD family remains unbounded, whereas NTS provides a strictly normalized measure essential for cross-context comparability.
>
> **Q5**:
> We thank the reviewer for this intriguing question. In our experiments, we did not observe significant negative values. Theoretically, a negative NTS implies an inverted topological hierarchy (where proximal pairs in Model A become distal in Model B). This is empirically rare, as trained networks inherently preserve shared semantic proximity, preventing complete structural inversion.
>
> **Q7**:
> We respectfully recommend NTS-E as the default choice. While NTS-M is more natural, NTS-E captures richer metric information (original edge distances), making it computationally simpler and empirically more discriminative.

---

### Author Response · Authors · 2025-11-25
**General review**

We thank the reviewers for their time and constructive feedback. We notice three common concerns regarding the novelty and theoretical positioning of our work. We wish to address these fundamental misunderstandings collectively here.
1. Clarification on **"Asymmetry"**：Some reviews suggested that RTD is already symmetric via averaging. We never claimed that $(RTD(A,B)+RTD(B,A))/2$is asymmetric. Rather, as explicitly stated in line 56(now in Line 64), our critique targets the brute-force averaging process itself. This heuristic approach masks the dramatic directional discrepancy between $RTD(A,B)$ and $RTD(B,A)$(shown as 'Min-Asym' in Table 2f (2)) and lacks theoretical interpretability. Our SRTD, resolves this by identifying the theoretical source of this asymmetry: we show that a large directional RTD often implies a small Max-RTD in the same direction (as shown in Table 2f (3)), and SRTD unifies these complementary views into a single, algebraically symmetric measure.

2. Clarification on **"Normalization"**: We apologize for any potential imprecision in our introduction.  It is correct that RTD uses 90th-percentile normalization; indeed, our SRTD and SRTD-lite also employ this technique in their calculations. Our true intent in Line 62 (now in Line 70)was to express that the output values of RTD can span an arbitrary positive range, lacking a suitable reference, and are significantly influenced by the scale of the point clouds (e.g., results for 100 points will almost certainly be much smaller than for 1000 points). This hinders cross-experiment interpretation and direct comparison. In contrast, our NTS is naturally constrained within the [-1, 1] range. This resolves the scaling issues inherent to divergence measures, rendering NTS a tool with superior comparability and interpretability.

3. Contributions:We emphasize that our work comprises **two distinct contributions**:
- SRTD: This part perfects the existing RTD and RTD-lite theoretical framework. By solving the symmetry problem at the algebraic level, we improve interpretability and, crucially, reduce computational time by ~50% compared to the original RTD, The computation of SRTD_lite also shows a slight improvement compared to RTD-lite.
- NTS (High Originality): NTS is not an incremental extension of RTD. It completely abandons the core tool of the RTD family—the "cross-barcode" (comparison with auxiliary graphs). Instead, NTS introduces a novel paradigm: comparing the merge order of 0-dimensional homology via rank correlation. It is a fundamentally new tool and, to the best of our knowledge, the only normalized topological measure. It addresses the scale alignment problem and decouples the score from point cloud size at the source, eliminating the need for empirical heuristics like 90th-percentile normalization — a method that, while capable of mitigating scale discrepancies, remains an imprecise workaround.

---

### Note · Authors · 2025-12-04

I have read and agree with the venue's withdrawal policy on behalf of myself and my co-authors.